# Photodynamic Therapy for Glioblastoma: Illuminating the Path toward Clinical Applicability

**DOI:** 10.3390/cancers15133427

**Published:** 2023-06-30

**Authors:** Debarati Bhanja, Hannah Wilding, Angel Baroz, Mara Trifoi, Ganesh Shenoy, Becky Slagle-Webb, Daniel Hayes, Yasaman Soudagar, James Connor, Alireza Mansouri

**Affiliations:** 1Department of Neurosurgery, Penn State College of Medicine, Hershey, PA 17033, USA; dbhanja@pennstatehealth.psu.edu (D.B.); hwilding@pennstatehealth.psu.edu (H.W.); abaroz@pennstatehealth.psu.edu (A.B.); mtrifoi@pennstatehealth.psu.edu (M.T.); gshenoy@pennstatehealth.psu.edu (G.S.); bwebb@pennstatehealth.psu.edu (B.S.-W.); jconnor@pennstatehealth.psu.edu (J.C.); 2Department of Biomedical Engineering, Pennsylvania State University, State College, PA 16801, USA; djh195@psu.edu; 3Neurescence Inc., Toronto, ON M5R 2Y9, Canada; yasaman.soudagar@bruker.com; 4Penn State Cancer Institute, Penn State Health, Hershey, PA 17033, USA

**Keywords:** glioblastoma, photodynamic therapy, photosensitizer, aminolaevulinic acid, pre-clinical, clinical trials

## Abstract

**Simple Summary:**

Glioblastoma (GBM) is the most common adult brain cancer. Despite extensive treatment protocols, all glioblastomas are eventually fatal. Photodynamic therapy (PDT) is a light-based treatment method, which offers delivery of anti-cancer treatment to focal areas, thereby limiting side effects. As PDT has become an attractive option to target glioblastoma cells, this review summarizes such experimental efforts. The aims of this review were to discuss both the potential and shortcomings of current PDT strategies, analyze the challenges which currently prevent PDT from being a viable treatment for GBM, and highlight novel investigations of this therapeutic option. The review concludes with a commentary on clinical trials currently furthering the field of PDT for GBM. Ultimately, through addressing barriers and proposing solutions, this review provides a path for optimizing PDT as a revolutionary treatment for GBM.

**Abstract:**

Glioblastoma (GBM) is the most common adult brain cancer. Despite extensive treatment protocols comprised of maximal surgical resection and adjuvant chemo–radiation, all glioblastomas recur and are eventually fatal. Emerging as a novel investigation for GBM treatment, photodynamic therapy (PDT) is a light-based modality that offers spatially and temporally specific delivery of anti-cancer therapy with limited systemic toxicity, making it an attractive option to target GBM cells remaining beyond the margins of surgical resection. Prior PDT approaches in GBM have been predominantly based on 5-aminolevulinic acid (5-ALA), a systemically administered drug that is metabolized only in cancer cells, prompting the release of reactive oxygen species (ROS), inducing tumor cell death via apoptosis. Hence, this review sets out to provide an overview of current PDT strategies, specifically addressing both the potential and shortcomings of 5-ALA as the most implemented photosensitizer. Subsequently, the challenges that impede the clinical translation of PDT are thoroughly analyzed, considering relevant gaps in the current PDT literature, such as variable uptake of 5-ALA by tumor cells, insufficient tissue penetrance of visible light, and poor oxygen recovery in 5-ALA-based PDT. Finally, novel investigations with the potential to improve the clinical applicability of PDT are highlighted, including longitudinal PDT delivery, photoimmunotherapy, nanoparticle-linked photosensitizers, and near-infrared radiation. The review concludes with commentary on clinical trials currently furthering the field of PDT for GBM. Ultimately, through addressing barriers to clinical translation of PDT and proposing solutions, this review provides a path for optimizing PDT as a paradigm-shifting treatment for GBM.

## 1. Introduction

Glioblastoma (GBM) is the most common adult brain cancer [1,2,3,4]. Despite maximal surgical resection and adjuvant chemo–radiation, all tumors eventually recur, and median survival is 14–18 months [5]. Major challenges in the clinical management of GBM include its diffusely infiltrative nature, difficulty developing therapeutic agents that adequately penetrate the blood-brain barrier (BBB), and risks associated with tumor resection adjacent to eloquent brain areas [6,7].

Photodynamic therapy (PDT) is a light-based treatment modality involving the administration of a photosensitizer to elicit a target response following photoactivation. Most PDT approaches involve the production of reactive oxygen species (ROS) to induce tumor cell death [8]. In theory, PDT can be a suitable option in GBM, addressing tumor cells that remain following maximal safe resection. Several strategies to increase the clinical applicability of PDT in GBM have thus far been attempted and, although promising, limitations in design and clinical feasibility have hindered progress.

In this review, we provide an overview of PDT principles and broadly summarize the ongoing PDT research efforts for GBM, with a focus on 5-aminolevulinic acid (5-ALA) as the most implemented photosensitizer. Analyzing challenges impeding clinical translation (Figure 1), we propose strategies for the improvement of scientific approach and research methodology specific to PDT in GBM. We conclude by highlighting promising new frontiers, beyond 5-ALA (Figure 2). The proposed optimization and enhancement of PDT could expand the armamentarium of adjuvant therapies in GBM, hinging on the consideration of the downstream cellular, metabolic, and genomic impact of PDT, as well as the emphasis on long-term clinical feasibility and applicability at the very infancy of drug and device design.

## 2. Photodynamic Therapy Principles: The Math and Biology

The governing variable of energy delivery in PDT is fluence rate, defined as $Fluence\;Rate=\frac{\mathrm{Joules}}{\mathrm{Seconds}*{\mathrm{cm}}^{2}}$. Optical power is the amount of energy deposited in tissue per unit of time (joules per second, or watts), and power density is the photon delivery unit expressed by watts per square centimeter (W/cm^2^) [9]. Optimizing these variables is crucial to PDT efficacy and clinical translatability.

PDT photosensitizers are broadly classified as either non-porphyrins or porphyrins, with porphyrins further classified as either first, second, or third generation [10]. Photosensitizers have evolved across generations, with their development increasingly focused on a profile of precise chemical composition and enhanced tumor target specificity [8,11]. To be sure, 5-ALA, a second-generation porphyrin and a precursor of protoporphyrin IX (PpIX), has garnered interest as a favorable photosensitizer in GBM, as it has already been FDA-approved for fluorescence-guided surgery (FGS) [8]. While not intrinsically fluorescent, 5-ALA is enzymatically converted to PpIX, a photosensitizer that can also be endogenously produced, by the heme biosynthesis pathway [12,13]. Exogenous delivery of ALA bypasses its bioproduction from glycine and succinyl-CoA via ALA synthase, the rate-limiting step of the synthesis, leading to significantly higher levels of PpIX than endogenous production; this process generates optically quantifiable in vivo levels of PpIX.

The advantages of 5-ALA over other photosensitizers are compelling. Numerous studies have highlighted the efficacy of 5-ALA-based PDT for GBM, with cytotoxicity levels as high as 80% in vitro and significant necrosis of tumor tissue in rat models [14,15]. Additionally, when 5-ALA is exogenously delivered, it leads to preferential production and accumulation of PpIX in cancer cells due to their high metabolism and downregulated activity of ferrochelatase, which converts PpIX into heme, thereby giving 5-ALA high tumor specificity [12]. Additionally, due to its current use in FGS, the safety profile of 5-ALA is well understood.

Yet, while 5-ALA PDT has been extensively explored in both preclinical and clinical studies, numerous challenges limit this potential treatment’s widespread applicability. Understanding the barriers to effective clinical translation begins with a critical evaluation of the 5-ALA PDT preclinical data. In vitro studies demonstrate mechanistic pitfalls, including variable cellular uptake of 5-ALA and limited consideration of the need for optimal oxygen recovery for ROS production. Dose delivery strategies that have shown the greatest efficacy in vitro have been minimally translated to animal models. Finally, an appraisal of the in vivo literature shows discrepancies in fluence rates and limited validation in putative translational models, which are essential for effective clinical translation. 

### 2.1. Variable Uptake of 5-ALA by Tumor Cells

To be sure, 5-ALA-based PDT is stringently dependent on the uptake and metabolic processing of 5-ALA by cancer cells. The mechanism of uptake involves passive diffusion as well as active transport through specific transporters, including ABC receptors [16]. Cellular subtype has also been implicated in varying degrees of uptake [8]. For example, the efficacy of 5-ALA uptake is much greater in bulk tumor cells in high-grade gliomas (HGGs) than in glioma stem cells (GSCs) [8] and lower-grade gliomas [17,18,19,20]. Given that tumor residual following surgery and adjuvant treatment is predominantly comprised of treatment-resistant GSC clones, conventional approaches to single-session intraoperative PDT may not address this vital therapeutic gap [18,19].

The combination of 5-ALA with other compounds that may improve the latter’s uptake has been attempted. Iron chelators, such as deferoxamine and CP94, and ATP-binding cassette transporter inhibitors, blocking channels including ABCG2 and ABCG6, have shown increased PpIX levels in tumor cells when utilized as adjuvants, especially in HGG [18,21,22]. Specifically, GSCs treated in vitro with 5-ALA in the absence and presence of an iron chelator, and the mean percentage of GSCs exhibiting PpIX fluorescence increased from 40.2% to 84.3% (*p* < 0.01) [18], respectively. Other methods include pharmacologic inhibition of ABCG2, which increased the mean PpIX fluorescence at least two-fold in sU251MG-V cells compared to controls (*p* < 0.05) [23]. Yet, the application of these adjuvant strategies in the clinical setting has been lacking due to a limited number of in vivo trials.

### 2.2. Visible Light Has Weak Tissue Penetration

Within the wavelength spectrum of visible light (380–800 nm), tissue penetration is only 8–12 mm [24]. Photoactivation of 5-ALA occurs at 635 nm, which can only penetrate 1–5 mm [25]. This challenge is even more prominent at concealed tumor locations of the resection cavity or tumor margins. Thus, adequate depth for tumor cytotoxicity is seldomly reached and most of the light energy is dispersed within the first 5 mm [6]. Innovative proposals to address this challenge have included the development of nanotechnology-based photosensitizers, new laser light sources, and optical fiber-guided laser transmission systems [26]. Given their low light transmission loss, improved target precision, and greater proximity of the light source to the tumor, optical fibers are key components of most adaptive light delivery devices [27] and demonstrate the potential for increased therapeutic depth of PDT. Other forms of novel light delivery for PDT include near-infrared (NIR) upconversion nanoparticles (UCNPs) [28] and bioluminescence [29]. Hypothetically, by converting NIR to visible light, acting as wireless transducers for PDT [28], the emission spectrum of the UCNPs can be matched with the absorption spectrum of 5-ALA, activating this photosensitizer and tumor cell death at deeper-seated brain areas. An in vivo mouse study performed by Teh et al. tested this concept using biocompatible UCNP implants, demonstrating shrinkage of tumor in PDT-treated mice [28]. 

### 2.3. Poor Oxygen Recovery in 5-ALA-Based PDT

Tumor hypoxia may be exacerbated by PDT, perhaps directly as a consequence of oxygen consumption to produce ROS, or indirectly, due to tumor vasculature blood flow stasis from the generated heat [30]. Diminished oxygen supply becomes a large barrier to the efficacy of 5-ALA-based PDT, preventing it from further generating cytotoxic ROS, thus inhibiting its full therapeutic potential. Direct vessel injury has also been reported [31]. Microvascular damage leads to platelet aggregation and decreased blood flow velocity and stasis. In vitro studies have investigated photosensitizers coupled with oxygen molecules as a potential method to sustain oxygen recovery [32,33]. For example, Cheng et al. loaded perfluorocarbon nanodroplets into photosensitizers to deliver oxygen self-enriched PDT, significantly decreasing tumor growth in breast and colon cancer cells. Such adjuncts have yet to be investigated in gliomas. 

Other efforts have focused on measuring real-time oxygen regeneration to adjust PDT settings as a method to prevent treatment-induced hypoxia [34,35,36]. The measurement of mitochondrial oxygen availability has been successfully utilized as a corollary to the total oxygen content in the tumor microenvironment (TME) and adjacent intravascular space. Ubbink et al. used COMET (cellular oxygen measurement device) to derive mitochondrial oxygen tension and oxygen disappearance rates before and after PDT, which in turn informed the timing and fluence rate of the subsequent PDT dose [37]. This dosimetric technique, while used to explore skin cancer [37], has yet to be applied to intracranial tumors owing to the device’s large size and implantation barriers.

### 2.4. Low and Steady Wins the Race?

While in theory, a high fluence rate may facilitate more efficient photoactivation, tissue damage, rapid oxygen depletion, and photobleaching are potential concerns. In some studies, low-fluence PDT has been shown to induce greater tumor cytotoxicity than high-fluence PDT, likely due to better oxygen replenishment and decreased tissue injury. Busch et al. showed poor oxygen recovery, particularly in tumor margins after high (78 mW/cm^2^) versus low (38 mW/cm^2^) power density PDT in a radiation-induced fibrosarcoma murine model [38]. Madsen et al. showed that lower fluence rates (<50 mW/cm^2^) produced a significantly greater cytotoxic effect on human glioma spheroids than high fluence rates (i.e., 150 and 200 mW/cm^2^) [39].

Photobleaching occurs from the biochemical decay of PpIX, and this can be accelerated by high optical powers. Belykh et al. demonstrated that fluorescence intensity and decay were highly dependent on fluence rates in in vivo malignant glioma models when comparing light power densities ranging from 5 to 21 mW/cm^2^ [40].

High fluence PDT is also associated with destructive off-target effects. Chen et al. calculated a “damage threshold”, which establishes the magnitude of fluence that is associated with injury to surrounding healthy brain tissue [41]. Fluence rates below this “damage threshold” produce a safer, more targeted response, whereas fluence rates above this threshold are more likely to precipitate secondary damage to normal brain tissue [42]. Unlike low fluence, which induces cytotoxicity through apoptosis, high fluence is thought to induce necrosis, affecting not only tumor tissue but also surrounding normal cells [43].

Based on the heterogeneity of GBM tumor cells and the presumptive need for a lower fluence rate, a longitudinal, low fluence PDT strategy may be necessary. In the preclinical setting, longitudinal PDT has been shown to be more efficacious in GBM, as evidenced by a landmark study from Hirschberg et al. Rat glioma spheroids, both in vitro and orthotopically implanted in vivo, treated with low-dose longitudinal PDT at 3-week-long treatment intervals inhibited spheroid growth better than single-treatment regimens [44]. Madsen et al. similarly found lower survival fractions in glioma spheroids treated with multiple PDT sessions (12 J, 12 J, and 25 J) compared to single session PDT with 12 J or 25 J alone. However, translating longitudinal 5-ALA PDT to the clinical setting poses a technical challenge. In addition to the obvious barriers imposed by the skull, the depth of light penetration is another concern hindering the development of extracranial, non-invasive devices for light delivery. To overcome this challenge, a chronically implanted light delivery apparatus may be necessary. 

## 3. Discrepancies in In Vivo 5-ALA Research

To understand the translational barriers, we conducted a systematic review focused on the in vivo PDT landscape. We searched PubMed/MEDLINE database with the following terms: “photodynamic therapy AND (“5-ALA” OR “ALA” OR “PpIX”) AND (“in vivo”) AND (“glioma” OR “glioblastoma” OR “astrocytoma”).” A total of 20 studies were populated. Inclusion criteria were the following: in vivo studies, 5-ALA photosensitizer utilized, and glioma pathology tested. Exclusion criteria were the following: review papers and extra-cranial pathology. Using these inclusion and exclusion criteria, we identified eight studies reporting on in vivo 5-ALA PDT for murine glioma models (Table 1) [14,15,43,44,45,46,47,48]. While the evidence for treatment efficacy was compelling, common limitations seen among studies included: (1) discrepant power densities and optimal fluence rates for balancing safety and efficacy; (2) limited mechanistic understanding of PDT impact on the immune response and the TME; and (3) absence of validation of optimal PDT settings in multiple cell lines or animal models.

One of the chief discrepancies in PDT in vivo research is the wide variation in power densities and fluence rates. For example, reportedly efficacious power densities have ranged from 0.5 mW/cm^2^ to 100 mW/cm^2^ [15,43]. Several studies lacked effective safety control groups; only three measured frequencies of hemorrhage, elevated intracranial pressure (ICP), and off-target cortical damage in response to treatment [14,45,48]. When safety endpoints were measured, it was seen that high power densities were associated with increased hemorrhage and elevated ICP compared to low power density treatment [14]. Only three studies used “light probe only” controls to isolate cortical injury induced by probe implantation alone. Finally, only three studies investigated the efficacy of low-dose longitudinal PDT delivery [14,43,44], and found increased treatment benefits compared to high-dose single-session PDT delivery in all three studies.

Only one of seven studies investigated the impact of PDT on the immune response and TME [46], and found increased CD8+ T-cell infiltration of the tumor in 5-ALA-Rutherrin coupled PDT compared to 5-ALA PDT alone. Considering that immune evasion and an immunologically cold TME are hallmarks of GBM tumorigenesis, this is a distinct and necessary step toward clinical translation that is broadly understated in the PDT literature [49].

While there is preliminary evidence to suggest PDT has an immune-modulating effect on GBM TME, further research with reliable preclinical models is necessary [50,51]. Syngeneic murine models are now considered the gold standard for immune-related studies [52]. In parallel with mechanistic insight gained from patient-derived xenografts, which retain the histological and genetic features of human GBM, this knowledge can inform future translational steps. Two studies identified in our systematic review used syngeneic mouse models [46,47]. None validated findings in external cohorts or analyzed patient-derived GBM xenografts. Murine models were mostly immunosuppressed CDF Fischer (2) or Wistar (2) rats with various orthotopically injected glioma cell lines. Studies also had low sample numbers, with the maximum number of animals in any given cohort limited to nine rats (one study), and all seven remaining studies having even fewer animals in each group. These limitations pose major challenges for translation to clinical studies. 

## 4. The Path Ahead

### 4.1. Longitudinal PDT Delivery

Innovations to investigate longitudinal PDT delivery have gathered momentum. Davies et al. designed a tetherless, light-weight, light-emitting diode (LED)-based fiber, which they implanted in the brains of 27 tumor-burdened rats at a depth of 2–2.5 mm, for longitudinal low fluence PDT [43]. The implantable prototype was well-tolerated, and the four rats in the longitudinal PDT group successfully endured the device over four days of the study. The longitudinal group also showed the greatest efficacy in tumor volume reduction (bioluminescent imaging signal), at a fluence rate of 0.5 mW/cm^2^ at 635 nm for four days. Additionally, animals in this treatment group had no tumor regrowth at 26 days from tumor implantation. However, the authors reported a significant challenge in designing and surgically fixing a durable, stereotactic device. Their final design used a cylindrical plastic adapter fixed with bone cement to maintain the fiber’s precise location [43]. Long-term durability still needs to be explored and must be considered in future feasibility studies, especially when translating the design to large animal models and human studies. 

An alternative approach to increase the efficiency of treatment delivery is the use of multiprobe apparatuses, coupled with imaging modalities for real-time tumor monitoring, to facilitate rigorous, large-scale animal studies. Multiple probes can simultaneously deliver light to both tumor-implanted and normal brain regions in several animals, enabling assessment of PDT impact on TME and normal brain in the same animal. It would also reduce variance in experimental conditions and increase efficiency by allowing simultaneous assessment of more than one animal at a time. Imaging modalities would allow for the measurement of autofluorescence to confirm photoactivation and rule out photobleaching.

Existing devices in other neuroscience applications successfully capitalize on these functionalities. Neuresence, for example, has developed the Chromatone^TM^ system. Chromatone^TM^ is an optical multiscope equipped with four flexible and lightweight, MRI-compatible probes, each delivering three different light wavelengths for simultaneous multicolor imaging and optogenetics stimulation of deep-seated neurons in multiple brain regions, with spatial and temporal resolution suitable for both real-time and longitudinal studies. It has successfully been implemented in in vivo experiments requiring multi-regional light delivery and functional imaging of neuronal activity. The surgical protocol has been validated in freely behaving rodents. The potential application of Chromatone^TM^ in PDT would involve the combination of photo-activation and fluorescent imaging for a fortified anti-tumor therapy. 

### 4.2. Harnessing Immunologic Response in PDT for GBM

A new frontier in PDT GBM research is coupled photo-immunotherapy (PIT), where a photosensitizer is conjugated to a highly specific monoclonal antibody (mAb) [53]. Light activation of the photosensitizer releases the mAb within the TME, thereby recruiting cells to trigger immunogenic cell death. Macynska et al. explored PIT/PDT using an IR700-labeled EGFR-specific affibody molecule in a syngeneic GBM murine model [54] and demonstrated that this PIT approach induced damage-associated molecular patterns and maturation of dendritic cells, as well as induced a T-cell response, all of which contributed to immunogenic cell death and therapeutic efficacy [54]. These results encourage further research to harness immunologic activation against GBM using PDT/PIT.

### 4.3. Nanoparticle-Linked miRNA Photosensitizers

A novel PDT strategy is to use nanoparticle-linked microRNAs (NP-miRNAs) as third-generation photosensitizers to release tumor suppressive miRNA mimics specifically within the GBM tumor environment following photoactivation. These mimics can diffuse beyond the site of release with enhanced permeation and retention in tumor cells, potentially increasing the therapeutic depth and effect. Compared to normal neural stem cells, GSCs express miRNAs known to influence tumorigenesis through effects on cell cycle regulatory proteins, cell differentiation or growth, and apoptosis [55]. Restoring anti-oncogenic miRNAs holds therapeutic potential.

Plasmonic gold–silver–gold core–shell–shell nanoparticles (NP) linked to miRNA serve as vectors for selective delivery, as the miRNA is photothermally cleaved and released from the nanoparticle only at the site of photoactivation with near-infrared (NIR) irradiation [56]. Liu et al. applied this strategy to administer exogenous miR-148b to transgenic mice with dermatological malignancies driven by a GTPase oncogene, HRas, resulting in efficient and safe tumor regression via apoptosis [57]. This study’s approach avoided common concerns with in vivo tumor-specific treatment, including cytotoxicity to adjacent keratinocytes, endosome escape of miRNA within the cytosol, and adequate spatiotemporal control. With our expanding knowledge of the unique role miRNAs play in glioma initiation and progression, we now have a novel opportunity for PDT strategies tailored for patients with GBM.

### 4.4. NIR Light Delivery

In contrast to the limited tissue penetration of visible light, the spectrum on which the photoactivating wavelength of 5-ALA occurs (635 nm), near-infrared radiation (NIR [800–2500 nm]) can penetrate an impressive 3 cm through craniofacial skin and bone structures [58]. Activation via visible light is a major limitation of 5-ALA as a photosensitizer for GBM since deeper areas of disease within the brain cannot be reached with such minimal tissue penetration. Strategies to overcome these limitations with 5-ALA may include multiple PDT probes, repeated treatments, and other innovative strategies for light delivery that are beyond the scope of this review. One study has presented the experimental protocol of utilizing a balloon filled with a diffusing solution attached to a trocar, paired with a fiber guide into which a cylindrical light can be placed [59]. This method can deliver PDT intraoperatively, post GBM resection, as the balloon is placed and filled with fluid within the resection cavity, then illuminated. This device is being further investigated in the intraoperative photodynamic therapy of GBM (INDYGO) clinical trials [60].

Alternatively, photosensitizers activated by light wavelengths with deeper tissue penetration would lessen the need for such methods of unique light delivery. In fact, therapies capitalizing on upconversion nanoparticle activation via NIR light are already being investigated for non-invasive deep brain stimulation of neurological disorders [61]. NIR photons also diminish both phototoxicity and background autofluorescence, leading to improved bioimaging, when compared to traditional fluorescence with visible light [62]. Activated within the NIR spectrum, nanoparticle-linked miRNA mimics facilitate predictable photoresponse by employing a thermally-labile linker to selectively release miRNA upon photo-irradiation at the particle’s respective plasmon wavelength [63]. Such characteristics translate to lower fluence requirements and more discrete wavelengths for photoactivation. Together, these features optimize the specificity of a miRNA-based PDT in GBM treatment delivery sites, providing a spatiotemporal benefit.

### 4.5. Other Nanoparticle-Linked Photosensitizers

Other in vitro PDT approaches have seen success utilizing nanoparticles. One recent study published in March 2023 utilized AGuIX^®^-design nanoparticles with an MRI contrast agent, porphyrin photosensitizer, and KDKPPR peptide ligand to investigate the influence of macrophage neuropilin-1 (NRP-1) protein expression, a protein known to impact GBM immune response and progression, on the uptake of nanoparticles [64]. Medium secreted by U87-MG tumor cells with or without PDT was incubated with THP-1 macrophages. This analysis found a three-fold increase in KDKPPR-functionalized nanoparticle uptake by M2 compared to M1 macrophages after 24 h of incubation, corresponding with three times more NRP-1 expression in M2 vs. M1 macrophages, assessed by mean fluorescence intensity values. The M0 and M2 macrophage populations also had significantly lower survival with both nanoparticle exposure and light compared to only light exposure, corresponding with a higher gene expression of NRP-1 in these phenotypes. Cell index evolution techniques were utilized to identify that the post-PDT U87 cell secretome preferentially polarized macrophages to the M1 phenotype. This study concluded that their nanoparticle PDT methodology promotes macrophage polarization toward M1 cells, which exhibit proinflammatory and antitumor properties, and targets M2 cells for destruction, which exhibit pro-tumor properties; hence, this method should be further tested with current interstitial PDT investigational efforts, demonstrating the potential to optimize inflammatory and vascular PDT responses.

Another study by Caverzán et al. investigated metronomic PDT with conjugated polymer nanoparticles (CPNs) in GBM, in attempts to overcome treatment resistance which occurs due to oxygen consumption during high fluence rate PDT [65]. Utilizing U87-MG, T98G, and M059K cell lines, the study compared metronomic PDT (mPDT) to high irradiance conventional PDT (cPDT) utilizing CPNs as photosensitizers; in vitro, they found that mPDT demonstrated more efficacious cell death, polarization of macrophages toward an antitumoral phenotype, and lower activity of molecular pathways corresponding with PDT resistance. When subsequently investigated in a heterotopic mouse model with U87-MG injected subcutaneously, results persisted, demonstrating apoptosis induction and tumor growth inhibition.

Other studies also suggest incorporating nanoparticle-based PDT to augment systemic therapies. For example, Pellosi et al. investigated nanoparticles delivering verteporfin as adjuvant therapy to temozolomide (TMZ) in vitro, concluding that this combination therapy acts synergistically to enhance antiproliferative effects and negate cross-resistance, thereby offering the potential to improve TMZ-based GBM treatment [66]. Another study compared anticancer activity through PDT with CPNs in three GBM cell lines with different redox statuses. The goal of their study was to consider methods that might optimize nanoparticle-based PDT for the profound tumor heterogeneity across GBM cells, concluding that cell-specific antioxidant enzymes vary across GBM cells and should be considered in creating selective and novel nanoparticle PDT treatments.

### 4.6. Phytocompound Photosensitizers

Phytocompounds have been investigated as photosensitizers with a low-toxicity profile; they are known to have a role in self-renewing signaling pathways including Hedgehog, Wnt/β-catenin, and Notch [67]. The literature has reported that curcumin can halt glioma cell growth by inhibiting the Sonic Hedgehog/glioma-associated oncogene homolog 1 (SHH/GLI1) pathway [68]. Other studies have reported its capability of targeting glioma-initiating cells in vitro and subsequently initiating autophagy and suppressing tumor formation in mouse models [69,70]. Curcumin has also been investigated as a photosensitizer in PDT; Kielbik et al. incubated SNB-19 cells for 2 and 24 h with 5–200 mM of curcumin, then irradiated the cells with blue light (6 J/cm^2^) and compared them to unirradiated controls [71]. At 2 h, the EC_50_ of curcumin was 6.3 times lower in cells with irradiation compared to those without; over 90% of the cells which had received irradiation underwent apoptosis.

Carriero et al. investigated a PDT method using berberine (BBR) as a photosensitizer. They investigated human astrocytoma cell lines T98G, U87-MG, U373-MG, and U138-MG in vitro, splitting cells into four study groups; group 1 was the control untreated, group 2 was control treated with only LED light, group 3 was treated with only BBR, and group 4 was treated with BBR and LED light [72]. Their data demonstrated in vitro evidence of BBR-induced apoptosis of the T98G cell line through immense ROS production, mitochondrial depolarization, and resulting caspase activation. Such apoptotic features were amplified in T98G cells post-irradiation, while the LED-only control treatment was not effective. Hence, phytocompounds demonstrate promise as photosensitizers for PDT for GBM, and the methodology detailed warrants further investigation.

## 5. Current Clinical Trials

The above propositions should be further investigated and pursued to improve the efficacy, specificity, and reliability of PDT, such that it can eventually become a mainstay in the treatment of GBM. Yet, many of these hypotheses are novel and at the in vitro phase of experimentation, still requiring many iterations of rigorous testing and proofs of concept prior to clinical translatability. Numerous clinical trials, the following of which are currently recruiting, are actively investigating the potential of PDT as a paradigm-shifting GBM treatment. A trial in Germany is exploring stereotactic biopsy followed by stereotactic 5-ALA-based PDT (NCT04469699); another German trial is evaluating the feasibility of 5-ALA for stereotactic interstitial PDT in a subset of adult patients with GBM (NCT03897491); and Dose Finding for Intraoperative Photodynamic Therapy of Glioblastoma (DOSINDYGO) (NCT04391062) is a dose-escalating extension of the intraoperative 5-ALA PDT INDYGO study, in part attempting to evaluate the impact of PDT dose on local GBM recurrence [73]. Hence, while not explicitly testing the novel methodologies of our proposed future directions for PDT, current clinical trials are still benefitting the PDT field by establishing feasibility, refining the workflow, and optimizing light delivery mechanisms. 

## 6. Conclusions

PDT has been extensively explored for GBM. However, various biological and translational barriers may limit the extent of clinical applicability. Future research initiatives should focus on promising frontiers including longitudinal delivery, photoimmunotherapy, and nanoparticle-linked miRNA photosensitizers for a more targeted anti-tumor approach, and NIR-based photoactivation for improved tissue penetration. Regardless of strategy, the ultimate translation to the clinical setting must always be considered, with the perspective of patients at the center of it all.

## Figures and Tables

**Figure 1 cancers-15-03427-f001:**
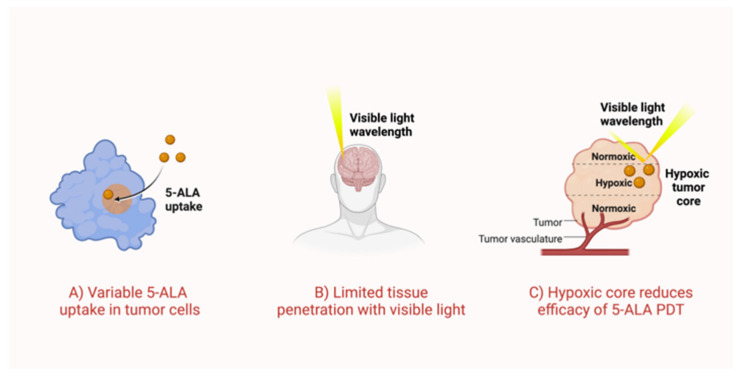
Shortcomings of 5-ALA PDT for glioblastoma. (**A**) There is variable uptake of 5-ALA in tumor cells due to surrounding microenvironment and physiological differences in tumor zones. Compounds such as iron chelators are hypothesized to improve uptake. (**B**) Excitation of 5-ALA is catalyzed by the visible light wavelength of 635 nm. However, visible light has poor tissue penetration, decreasing potential therapeutic depth of 5-ALA PDT. (**C**) The hypoxic core of a GBM tumor reduces the efficacy of 5-ALA PDT, as this therapy is dependent on oxygen availability.

**Figure 2 cancers-15-03427-f002:**
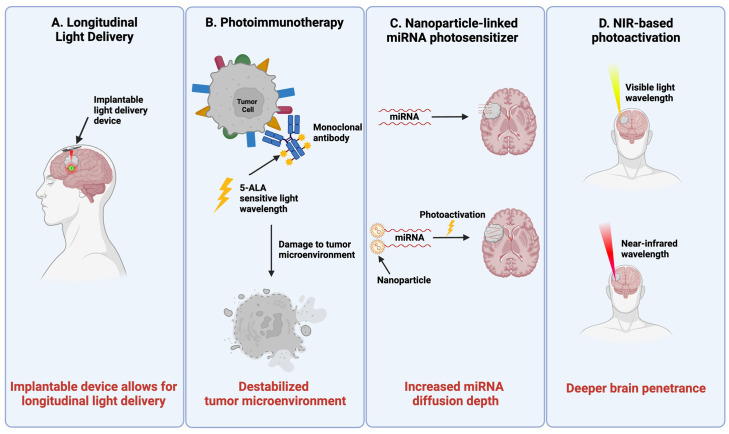
Key strategies to improve clinical translatability of PDT. (**A**) Longitudinal PDT delivery: implantable light delivery devices allow for longitudinal PDT therapy delivery. (**B**) Photoimmunotherapy: the addition of photoactivation to immunomodulating therapies can improve damage to the tumor microenvironment. (**C**) Nanoparticle-linked miRNA (NP-miRNAs) photosensitizers: NP-miRNAs allow for increased depth of diffusion of miRNA particles in comparison to miRNA molecules without nanoparticles. Additionally, these novel photosensitizers do not consume oxygen, thereby theoretically eliminating concerns of PDT-induced hypoxia. (**D**) Near-infrared (NIR) photoactivation: visible light allows for 1 cm of penetration through the human cranium, while NIR irradiation allows for up to 3 cm of light penetration leading to improved delivery at the tumor site.

**Table 1 cancers-15-03427-t001:** Highlights of published work on in vivo PDT.

Author, Year	Animal Model (# of Animals)	PDT Parameters	Outcome Variables	Main Results	Shortcomings
Hirschberg, 2006 [42]	BDIX rats, BT4C HGG spheroids (15)	Longitudinal; 7–30 mW; 10–30 min/week (×3)	-PpIX biodistribution -Overall survival -Necrosis	1. ↑ Overall survival in repetitive PDT compared to single session. 2. ↑ Necrosis in low fluence groups compared to high fluence.	-Did not control for probe or heat damage -Repeat surgeries and anesthesia for each PDT delivery = stress on animal
Davies, 2007 [41]	Fischer Rats, CNS-1 Astrocytoma (37)	Longitudinal; 0.5 mW/cm^2^; 24–96 h	-Tumor volume -Tumor regrowth -Necrosis -Apoptosis	Tumor volume reduction greater with 96 h vs. 24 h.	-Did not control for probe or heat-induced damage -Astrocytoma model, not GBM or high-grade -Duration not translatable
Tetard, 2016 [44]	Fox1 rnu/rnu rats, U87 GBM (22)	Fractionated and continuous; 4.8–30 mW; 120 s between 5 J and 21 J of delivery	-Necrosis -ICP -Hemorrhage	1. ↑ Necrosis in fractionated group compared to continuous. 2. ↑ ICP and hemorrhage in high fluence group.	-Single session PDT -Histological images provided do not compare normal brain to PDT treated area
Yi, 2015 [45]	Wistar rats, C6 glioma cells (24)	Single session; 100 mW/cm^2^; 60 min	-Tumor size -Tumor volume -Necrosis -Micro-vessel density (MVD) -Apoptosis	1. ↓ Tumor volume in PDT group compared to controls. 2. ↑ Necrosis in PDT group compared to controls. 3. ↓ MVD in PDT group compared to controls. 4. No difference in apoptosis between groups.	-Single session -Not an intracranial model (graft implanted in abdomen)
Munegowda, 2019 [46]	CDF Fischer rats, RG-2 cells (46)	Single session; 18 mW; 22 min	-Survival -Tumor volume -Intratumor edema -CD8 T-cell stain	Compared 5-ALA to Rutherrin photosensitizer: 1. ↑ Overall survival in R-ALA- and Rutherrin-treated rats compared to controls. Rutherrin increased survival more than 5-ALA. 2. ↓ Edema in Rutherrin group compared to 5-ALA. 3. ↑ CD8+ T-cell infiltration in Rutherrin compared to 5-ALA groups.	-Single session PDT -Did not compare 5-ALA to controls for edema and CD8+ T-cell infiltration -Small control cohort (n = 4) -Small PDT cohort (n = 6)
Olzowy, 2002 [43]	Wistar, C6 glioma cells (30)	Single session; 100 mW/cm^2^	-Cortical damage -Tumor size -Hemorrhage	1. No difference in cortical damage between PDT group and no irradiation group.	-Single session PDT -Histological images provided did not compare normal brain to PDT treated areas
Fisher, 2017 [47]	CDF Fischer rats, RG-2 cells (12)	Single session; 18 mW for 22 min	-Intratumor edema -Reactive gliosis -Survival	Compared 5-ALA PDT in hypothermic and normothermic conditions: 1. ↓ Edema in hypothermic conditions. 2. ↑ PpIX fluorescence in hypothermia conditions. 3. ↑ median overall survival in hypothermic conditions. 4. Increased cellular protection to normal brain structures in hypothermic conditions.	-Single session PDT -High variation in edema data
Fisher, 2019 [48]	GSC30 Rag2 -/- SCID rats, U87 cells (20)	Single session; 22.2 mW/cm^2^	-Hypoxia -Tumor blood flow -Survival -Hemorrhage	Compared 5-ALA PDT with and without lapatinib and lapatinib along: 1. ↑ Overall survival in PDT + lapatinib group. 2. No difference in hypoxia between groups. 3. ↓ Tumor blood flow in PDT + lapatinib group. 4. ↑ Edema in PDT + lapatinib group	-Single session PDT -results only applicable to EGFR sensitive tumors

↑: Increased; ↓: Decreased.

## Data Availability

The data presented in this study are available on request from the corresponding author.

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
