# Peer review of "Photodynamic Therapy for Glioblastoma: Illuminating the Path toward Clinical Applicability"

_cancers, 2023, doi:10.3390/cancers15133427_

Round 1

Reviewer 1 Report

Journal Cancers (ISSN 2072-6694)

Manuscript ID cancers-2465573

Type Review

Title Photodynamic therapy for glioblastoma: Illuminating the path toward clinical applicability

Authors Debarati Bhanja , Hannah Wilding , Angel Baroz , Mara Trifoi , Ganesh Shenoy , Becky Slagle-Webb , Daniel Hayes , Yasaman Soudagar , James Connor , Alireza Mansouri

The Authors summarized PDT technology within glial tumors with the highest malignancy as glioblastoma.

The topic is relevant considering the very poor prognosis for glioblastoma patients to date.

They mainly focused on clinically 5-ALA discussing benefits and technical and biological limits. As a novel and promising approach, Authors briefly discuss about nanoparticles as possible vehicles of photosensitizers. However, this innovative and ongoing approach is not enough explored by the Authors. I strongly suggest to increase this section, not only limiting to the reported miRNA, but reviewing different nanoparticles strategies employed in glioblastoma, of course these studies are mainly in vitro ones. I also believe that the general title adopted by Authors merits a wider exploitation of even different photosensitizers, not only 5-ALA, but for example including plant natural alkaloids as berberine and curcumine, etc.

In conclusion, even if the manuscript is well written and documented, I strongly suggest to increase the content as above reported.

Author Response

Thank you very much for your thoughtful review and raising these points. We have added subsections on additional nanoparticle and phytocompound photosensitizers in section 4.

Reviewer 2 Report

The review paper is short but abundant, and has a comprehensive overview of photodynamic therapy (using 5-ALA) for glioblastoma, it has a logical structure, I’d suggest its publication after the revisions:

1.      The unique advantages of 5-ALA out of other photosensitizers should be listed in a single paragraph.

2.      Here, PDT should be light:

3.      In Figure 2 (corresponding part 4), no method to solve the hypoxic issue is mentioned, can you supplement some solutions for this?

4.      In part 4.4, you mentioned “near-infrared radiation (NIR [800nm-2500nm]) can penetrate an impressive 3 cm through craniofacial skin and bone structures”, however, the absorbance of 5-ALA is 635 nm, not match, do you have any idea to solve this problem?

5.      What is PDL-506 in part 5?

6.      What does the number mean in the table? Is it a reference?

minor grammer problems

Author Response

  1. The unique advantages of 5-ALA out of other photosensitizers should be listed in a single paragraph.

Thank you for bringing this to our attention. We have added a paragraph in section 2 on the advantages of 5-ALA.

  1. Here, PDT should be light: --

Thank you – we corrected this in the figures.

  1. In Figure 2 (corresponding part 4), no method to solve the hypoxic issue is mentioned, can you supplement some solutions for this?

Thank you for your detailed review and helpful comments on our manuscript. This is a great point. The solution to PDT-induced hypoxia is inherent to using different nanoparticles, such as miRNA-linked nanoparticle photosensitizers. These photosensitizers do not consume oxygen to produce their anti-tumor effect, whereas 5-ALA consumes oxygen in the metabolism of 5-ALA into reactive oxygen species. To your point, we have added this clarifying explanation in the Figure 2 caption. This discussion can be found on page 4 and 5 of the manuscript.

  1. In part 4.4, you mentioned “near-infrared radiation (NIR [800nm-2500nm]) can penetrate an impressive 3 cm through craniofacial skin and bone structures”, however, the absorbance of 5-ALA is 635 nm, not match, do you have any idea to solve this problem?

Thank you very much for this comment. The NIR is mentioned as it is the wavelength which activates the nanoparticle miRNA photosensitizers, further optimizing these particles as potential photosensitizers due to their increased potential depth. Hence, this wavelength does not activate 5ALA, and minimal depth of penetration is a large disadvantage of 5-ALA. We have added this clarification in section 4.4, 4.5 and 4.6.

  1. What is PDL-506 in part 5?

Thank you for bringing this question. We have added clarity to the sentence.

  1. What does the number mean in the table? Is it a reference?

Thank you for raising this point. We added clarity to the table by including a separate column for author and citation, and then clarified the numbers correspond to numbers of animals used in the study.

Reviewer 3 Report

This manuscript is a review of the use of photodynamic therapy of glioblastoma in murine models using 5-ALA as a photosensitizer. 

1. The authors found only 8 in vivo studies murine studies using 5-ALA  PDT in glioblastoma (line 205-206). It would be helpful if the authors gave more detail regarding their literature search. For instance, it would be helpful to know what databases were searched and which keywords were used. It would also be helpful if the authors would include what inclusion/exclusion criteria were used in their literature search to determine relevance.

2. It would be helpful if the references summarized by table 1 (ref 41-48) were included in the table. Likewise, it is unclear why the first three table entries (presumably corresponding to references 41, 42, and 43) have other references (9, 11, and 12, respectively) also listed at the end of the “animal model, # of animals” column in their table entries.

Author Response

  1. The authors found only 8 in vivo studies murine studies using 5-ALA  PDT in glioblastoma (line 205-206). It would be helpful if the authors gave more detail regarding their literature search. For instance, it would be helpful to know what databases were searched and which keywords were used. It would also be helpful if the authors would include what inclusion/exclusion criteria were used in their literature search to determine relevance.

Thank you for this great point. We completely agree, and to provide more context, we have added the databases searched, the number of articles identified, and the inclusion/ exclusion criteria. This hopefully provides more context to the literature review.

  1. It would be helpful if the references summarized by table 1 (ref 41-48) were included in the table. Likewise, it is unclear why the first three table entries (presumably corresponding to references 41, 42, and 43) have other references (9, 11, and 12, respectively) also listed at the end of the “animal model, # of animals” column in their table entries.

Thank you for your detailed review of our manuscript and identifying this confusion in our text. We have added the correct references, as well as added a column in the table with the Author and year of publication. Using the table header, we have clarified that “# of animals” is listed in the parentheses ( ) within that column.

Reviewer 4 Report

This is a topical review that nicely sums up the current knowledge in this field.

Minor comments

line 15: delete the word paper

Table 1: Check references (style, numbers)

Author Response

line 15: delete the word paper

Thank you for your thoughtful review. Completed this request.

Table 1: Check references (style, numbers)

Completed this request.

Reviewer 5 Report

The review is comprehensive and cover the topic with a sufficient amount of  up-to date litterature. 

I so support the publication of the following article. 

The English can be improved but it is overall understandable and clear. 

Author Response

Thank you so much for your thoughtful review of our work.